# Wheat Oxylipins in Response to Aphids, CO_2_ and Nitrogen Regimes

**DOI:** 10.3390/molecules28104133

**Published:** 2023-05-16

**Authors:** Mari Merce Cascant-Vilaplana, Eduardo Viteritti, Víctor Sadras, Sonia Medina, María Puerto Sánchez-Iglesias, Camille Oger, Jean-Marie Galano, Thierry Durand, José Antonio Gabaldón, Julian Taylor, Federico Ferreres, Manuel Sergi, Angel Gil-Izquierdo

**Affiliations:** 1Research Group on Quality, Safety and Bioactivity of Plant Foods, Department of Food Science and Technology, CEBAS-CSIC, University Campus of Espinardo, Edif. 25, 30100 Murcia, Spain; 2Neonatal Research Group, Health Research Institute La Fe, 46026 Valencia, Spain; 3Faculty of Bioscience and Technology for Food, Agriculture and Environment, University of Teramo, Via Renato Balzarini 1, 64100 Teramo, Italy; 4South Australian Research and Development Institute, Adelaide, SA 5064, Australia; 5School of Agriculture, Food and Wine, The University of Adelaide, Adelaide, SA 5005, Australia; 6Institut des Biomolécules Max Mousseron (IBMM), Pôle Chimie Balard Recherche, UMR 5247, CNRS, University of Montpellier, ENSCM, 34090 Montpellier, France; 7Molecular Recognition and Encapsulation Research Group (REM), Health Sciences Department, Universidad Católica de Murcia (UCAM), Campus de los Jerónimos 135, 30107 Guadalupe, Spain

**Keywords:** wheat, aphids, nitrogen, CO_2_, plant oxylipins, phytoprostanes, phytofurans, oxidative stress, *Rhopalosiphum padi*, *Sitobion avenae*

## Abstract

Wheat is critical for food security, and is challenged by biotic stresses, chiefly aphids and the viruses they transmit. The objective of this study was to determine whether aphids feeding on wheat could trigger a defensive plant reaction to oxidative stress that involved plant oxylipins. Plants were grown in chambers with a factorial combination of two nitrogen rates (100% N vs. 20% N in Hoagland solution), and two concentrations of CO_2_ (400 vs. 700 ppm). The seedlings were challenged with *Rhopalosiphum padi* or *Sitobion avenae* for 8 h. Wheat leaves produced phytoprostanes (PhytoPs) of the F_1_ series, and three types of phytofurans (PhytoFs): *ent*-16(*RS*)-13-*epi*-ST-Δ^14^-9-PhytoF, *ent*-16(*RS*)-9-*epi*-ST-Δ^14^-10-PhytoF and *ent*-9(*RS*)-12-*epi*-ST-Δ^10^-13-PhytoF. The oxylipin levels varied with aphids, but not with other experimental sources of variation. Both *Rhopalosiphum padi* and *Sitobion avenae* reduced the concentrations of *ent*-16(*RS*)-13-*epi*-ST-Δ^14^-9-PhytoF and *ent*-16(*RS*)-9-*epi*-ST-Δ^14^-10-PhytoF in relation to controls, but had little or no effect on PhytoPs. Our results are consistent with aphids affecting the levels of PUFAs (oxylipin precursors), which decreased the levels of PhytoFs in wheat leaves. Therefore, PhytoFs could be postulated as an early indicator of aphid hosting for this plant species. This is the first report on the quantification of non-enzymatic PhytoFs and PhytoPs in wheat leaves in response to aphids.

## 1. Introduction

Biotic (insects, pathogens, weeds) and abiotic stresses (drought, salinity, extreme temperatures) compromise agricultural production [1,2,3]. Elevated [CO_2_], high temperatures and nitrogen deficiency can also influence the nutritional quality of plants and their secondary metabolite profiles [4,5].

Jasmonic acid (JA) is a lipid-derived plant hormone that is synthesized from α-linolenic acid (ALA; C18:3); it plays important roles, including in defense responses against biotic stresses [6]. Specifically, JA and its derivatives are polyunsaturated fatty acids (PUFAs) that are derived from cyclopentanones and belong to the family of oxidized lipids known collectively as oxylipins [7]. Many studies reported on the high bioactivity of JA and jasmonates in response to abiotic and biotic stresses, promoting protective mechanisms [4,8,9,10]. These compounds are produced by the enzymatic oxidation of α-linolenate. In the jasmonate pathway, catalyzed by lipoxygenases, α-linolenic acid (ALA; C18:3) is converted into 12, 13-(S)-epoxy-octadecanoic acid (12, 13-EOT), the first intermediate in the biosynthesis of JA [11]. Where reactive oxygen species (ROS) non-enzymatically catalyzes the oxidation of ALA in cell membranes, different phytoprostanes (PhytoPs) can form prostaglandins containing D_1_, E_1_, F_1_, A_1_, B_1_, L_1_ or the deoxy J_1_ ring system, as well as malondialdehyde [12,13,14,15,16,17,18]. Cuyamendous et al. [19] reported a new type of vegetable oxylipin resulting from a similar lipid oxidation reaction. This process is activated at an oxygen pressure higher than 21% that leads, after a first cyclization, to the generation of cycles containing tetrahydrofurans, the so-called phytofurans (PhytoFs) [17,19]. PhytoPs and PhytoFs are common in plants, and have been seen as biomarkers of oxidative stress in response to abiotic stress [20,21,22,23]. Oxidative stress is a complex, widespread chemical and physiological phenomenon in plants [24]. It develops as a result of the overproduction and accumulation of ROS [25]. PhytoPs and PhytoFs are both an oxidation index of plant lipids [26,27], and a biomarker for quality control of plant food products during manufacturing and storage [16]. Processing techniques, such as roasting or frying and storage at different temperatures, were found to affect the generation of PhytoPs in food products such as sea buckthorn, vegetables, olives, grape must, almond kernels, nuts and pistachios [20,28,29,30,31,32,33,34,35,36]. In cereals, PhytoPs and PhytoFs have been identified and quantified in rice bran and wheat flour [37] and also in peas [38].

The bird cherry-oat aphid, *Rhopalosiphum padi*, and the English grain aphid, *Sitobion avenue*, are major pests of wheat worldwide. With a focus on this wheat–aphid system, we reported the role of carbohydrates, with an emphasis on osmotic stress, on the fitness and behavior of these insects [5]. Using the same experimental setting, we investigated whether aphids trigger a defensive reaction from the plant in terms of oxidative stress related to the generation of oxylipins.

## 2. Results

### 2.1. Wheat Samples

A total of 29 samples of wheat were included in this study. Six of twenty-nine samples were not treated with aphids (controls), twelve were treated with *Sitobion avenae* and eleven with *Rhopalosiphum padi*. No significant differences between (i) control vs. *Rhopalosiphum padi* aphid; (ii) samples without aphid treatment vs. Sa aphid; (iii) *Rhopalosiphum padi* vs. *Sitobion avenae* aphids; and (iv) control vs. Aphids (*Rhopalosiphum padi* + *Sitobion avenae*) were found for the nitrogen and CO_2_ regimes. All of the groups were balanced; hence, they were unbiased for the effects of nitrogen and CO_2_ in the between-group comparisons.

### 2.2. Qualitative Profile of Phytoprostanes and Phytofurans in Wheat Leaves

Of the 10 plant oxylipins analyzed, 4 PhytoPs (9-F_1t_-PhytoP, 9-*epi*-9-F_1t_-PhytoP, *ent*-16-F_1t_-PhytoP + *ent*-16-*epi*-16-F_1t_-PhytoP) and 3 PhytoFs (*ent*-9(*RS*)-12-*epi*-ST-Δ^10^-13-PhytoF, *ent*-16(RS)-13-*epi*-ST-Δ^14^-9-PhytoF and *ent*-16(*RS*)-9-*epi*-ST-Δ^14^-10-PhytoF) were detected in wheat leaves, regardless of treatment application with N or CO_2_ regimes, and did not affect the qualitative profile of oxylipins in wheat leaves. The isomeric nature of *ent*-16-F_1t_-PhytoP and *ent*-16-*epi*-16-F_1t_-PhytoP precluded their separation via normal chromatographic techniques using a C_18_ column, so we quantified them together as in previous studies [12].

### 2.3. Phytoprostane and Phytofuran Content in Wheat before and after Aphid Treatment

PhytoPs and PhytoFs were determined in 29 wheat samples. Figure 1 shows the PCA plot based on the PhytoF and PhytoP levels of the three groups of wheat samples, including wheat samples without aphid hosting (control), wheat samples treated with *Rhopalosiphum padi* (Rp) and wheat samples treated with *Sitobion avenae* (Sa). The scores of PC1 vs. PC2 represent 89% of the explained variance, reflecting a clear trend with the treatment in the direction of PC2 (Figure 1A). It can be seen that there are three different types of clusters, where each one represents the aphid hosting. Furthermore, a total PhytoP-pattern in the direction PC1 (Figure 1B) was observed, indicating that wheat sample leaves challenged with *Rhopalosiphum padi* had higher levels of total PhytoPs than their counterparts with *Sitobion avenae* and controls (Figure 1B). Figure 1C shows the loadings for PC1 vs. PC2 that highlight elevated levels of *ent*-16(*RS*)-13-*epi*-ST-Δ^14^-9-PhytoF and *ent*-16(*RS*)-9-*epi*-ST-Δ^14^-10-PhytoF in the controls.

Table 1 shows the concentration of the oxylipins found in this study. No significant differences in the levels of 9-*epi*-9-F_1t_-PhytoP, *ent*-16-*epi*-16-F_1t_-PhytoP+*ent*-16-F_1t_-PhytoP, *ent*-9(*RS*)-12-*epi*-ST-Δ^10^-13-PhytoF and total PhytoP were observed between the controls and leaf samples treated with *Sitobion avenae* or *Rhopalosiphum padi*. However, levels of 9-F_1t_-PhytoP, *ent*-16(*RS*)-13-*epi*-ST-Δ^14^-9-PhytoF, *ent*-16(*RS*)-9-*epi*-ST-Δ^14^-10-PhytoF and the sum of total PhytoF presented significant differences. Aphids affected the levels of *ent*-16(*RS*)-13-*epi*-ST-Δ^14^-9-PhytoF and *ent*-16(*RS*)-9-*epi*-ST-Δ^14^-10-PhytoF. Nitrogen and CO_2_ regimes did not affect the global levels of PhytoPs or PhytoFs, or the particular individual oxylipins that were evaluated (*p* > 0.05).

### 2.4. Effect of Aphids on Phytoprostane and Phytofuran Levels in Wheat Leaves

We developed four binary PLSDA models to assess PhytoP and PhytoF levels in wheat leaves: control vs. *Rhopalosiphum padi* (Figure 2A, top); control vs. Sa (Figure 2B, top); *Rhopalosiphum padi* vs. *Sitobion avenae* (Figure 2C, top); and control vs. aphids (*Rhopalosiphum padi* + *Sitobion avenae*) (Figure 2D, top). As can be seen in all four binary PLSDA models, there are two different types of clusters, which indicate the kinds of aphid hosting. Moreover, in order to test the significance of the models, the *p*-values were obtained through permutation testing, indicating that there were significant differences between the tested groups (*p* < 0.05). The ROC curves based on the results were obtained with leave-one-out cross-validation for the ability of PhytoPs and PhytoFs to distinguish between control vs. *Rhopalosiphum padi* aphid (Figure 2A, middle); control vs. *Sitobion avenae* aphid (Figure 2B, middle); *Rhopalosiphum padi* vs. *Sitobion avenae* aphids (Figure 2C, middle); and controls vs. aphids (*Rhopalosiphum padi* + *Sitobion avenae*) (Figure 2D, middle). The areas under the curves (AUCs) were calculated from the ROC curve cross validation classification model, and resulted in values higher than 0.8106. It should be noted that the maximum value of an AUC is 1, revealing that a model having a value closer to 1 is more reliable, while a value closer to 0 is poor in its performance. Therefore, the performance of the current models based on PhytoPs and PhytoFs were appreciable for discriminating between aphid hosting. Thus, analyte levels that were found in wheat with treatment were significantly different from the controls as well as between aphid groups, as confirmed by permutation testing (*p* < 0.05) in all four cases. The variable importance in projection scores versus regression vectors (Figure 2A–D, bottom) were used to measure the influence of each metabolite on the PLSDA models. The levels of *ent*-16(*RS*)-9-*epi*-ST-Δ^14^-10-PhytoF were higher in the controls than with aphids (Figure 2A,B,D, bottom), and levels of *ent*-16(*RS*)-9-*epi*-ST-Δ^14^-10-PhytoF were higher in *Sitobion avenae* than *Rhopalosiphum padi* (Figure 2C, bottom).

## 3. Discussion

Biotic and abiotic stresses induce morphological, physiological, biochemical and molecular changes that affect crop growth and yield [2]. The interest in determining PhytoP and PhytoF levels in wheat samples was two-fold: as indicators of oxidative stress, and their putative role in defense. Fatty acid desaturases (FADs) can modulate plants’ defenses to pathogens and insects [39]. PUFAs generated by FADs are precursors for multiple oxylipins that contribute to plant defense and developmental pathways in plants that vary with ontogeny and in response to pathogens and insects [40]. These stress responses usually include the production of specific oxylipins, which have many biological functions [7].

In this study, we determined PhytoPs and PhytoFs in wheat leaves with *Rhopalosiphum padi*, *Sitobion avenae* and in controls with no aphids. The usefulness of this research may be of interest from a physiological point of view on the behavior of the wheat plant against hosting aphids, and also in finding early markers of the invasion of these aphids that would be useful to apply measures to, in order to reduce this pest and minimize possible losses in the quality or production of wheat. We detected four PhytoPs (Table 1). The complete F series (9-F_1t_-PhytoP, 9-*epi*-9-F_1t_-PhytoP, *ent*-16-F_1t_-PhytoP + *ent*-16-*epi*-16-F_1t_-PhytoP) qualitatively coincided with those found in *Cucumis melo*, date trees (*Phoenix dactylifera*) and red and brown macroalgae (Table 2 and Appendix A) [41,42,43,44,45]. This qualitative presence extends to other tissues and species, such as the cotyledons, shells, the calyx of Chilean hazelnut (*Gevuina avellana*), *Passiflora tripartita* and *Passiflora edulis* and *Physalis peruviana*, as well as date tree skin, pits, pulp and clusters, and cocoa pod husks (Table 2 and Appendix A) [18,46,47,48]. Fruits showed a greater variety of PhytoPs in most cases, including cereals such as rice (eight PhytoPs), legumes, nuts, cocoa bean and coffee pulp (Table 2 and Appendix A) [22,23,28,29,37,38,49,50,51,52,53], and predominantly PhytoPs of the F_1_, L_1_ and B_1_ series with respect to only the presence of the F_1_ series in wheat leaves (Table 2 and Appendix A). Processed plant food provided the greatest variability in PhytoPs. This is because, in addition to the genetic and environmental sources of the plant phenotype, processing (milling, grinding, tempering) favors exposure to ROS, which leads to more types of PhytoPs [13,49]. As for PhytoFs, the presence of the three compounds detected in wheat in our study coincided with those found in *Cucumis melo leaves*, date tree leaves, three types of brown macroalgae (*Ectocarpus siliculosus*, *Laminaria digitate*, *Fucus spiralis*), Chilean hazelnut cotyledons, and in date tree skin, pits, clusters, pollen and pulp, as well as cocoa pod husks (Table 3 and Appendix A). These oxylipins varied more in our wheat study than in red macroalgae and one of the brown macroalgae species, which lacked *ent*-9(*RS*)-12-*epi*-ST-Δ^10^-13-PhytoF (Table 3 and Appendix A). The three types of PhytoFs from wheat leaves in our study were found in all fruits except in flax and chia seeds (Table 3 and Appendix A). In general, food processing was detrimental to PhytoFs, as only one or two types of PhytoFs were present compared to the three PhytoFs contributed by wheat leaves as a counterpoint to the increased PhytoPs (Table 3 and Appendix A).

The total PhytoPs in wheat leaves were between 0.005 and 0.02 µg/100 g f.w. for the controls, and increased up to between 1.2 and 0.03 µg/100 g f.w. in response to *Rhopalosiphum padi* and *Sitobion avenae*. This range of PhytoP concentrations in wheat leaves was similar to those found in the leaves of *Tilia cordata*, *Betula pendula*, *Lycopersicum sculentum*, *Salix alba* and *Rauvolfia serpentina*, and were lower than those found in *Nicotiana tabacum*, *Arabidopsis thaliana*, *Cucumis melo* and *Mentha piperita* (Table 2 and Appendix A).

Biotic stress is usually accompanied by oxidative stress and the overproduction of ROS and, consequently, PhytoPs and PhytoFs. PhytoFs are oxylipins that share a structural analogy with PhytoPs generated by non-enzymatic oxidative reactions as well, although higher oxygen pressures (>21%) tip the scale in favor of PhytoF syntheses [54].

Other studies have included some fatty acids as other candidate markers of oxidative stress from plant–pathogen interactions (candidates for indirect oxidative stress through fatty acid degradation) [55]. However, we found that PhytoFs could be an early indicator of aphid hosting. Wheat leaves in the control plants had higher concentrations of *ent*-16(*RS*)-13-*epi*-ST-Δ^14^-9-PhytoF and *ent*-16(*RS*)-9-*epi*-ST-Δ^14^-10-PhytoF than aphid-treated plants (metabolites of linolenic acid oxidation, which indicate direct oxidative stress) (see Table 1), with the *Rhopalosiphum padi* group having the lowest concentration levels. However, no significant differences in PhytoP content were found between groups, except for 9-F_1t_-PhytoP levels in samples treated with *Rhopalosiphum padi*, which were significantly higher than in samples without aphid treatment. Thus, the infestation of *Rhopalosiphum padi* and *Sitobion avenae* led to a significant decrease in PhytoP levels in wheat plants. There is little information about this class of oxylipins in relation to biotic stress. Chewing insects induce the release of linolenic acid from the lipids of the intracellular membrane [56]. Lipids are released from membranes, and function as signal molecules in the activation of plant defense responses such as oxylipin synthesis. Oxylipin biosynthesis is very dynamic, and takes place both in the constitutive state and in response to plant–pathogen interactions. Oxylipin signals play a role in a variety of signaling pathways, making them essential parts of the plant’s innate immune network [57]. We hypothesized that aphids decrease PUFA metabolism (particularly, ALA) in plants, and this reduction affects the production of PhytoFs and PhytoPs. Plants respond to abiotic and biotic stress through the adaptive remodeling of membrane fluidity and fatty acid composition [57,58,59]. Saturated vs. unsaturated lipid ratios play a crucial role in plant survival and stress tolerance [60]. In this experimental setting, we found that high levels of CO_2_ increased fructose and glucose concentrations, which are essential to buffer the growth requirement of wheat, and play an osmotic role in plant resistance to aphids [5]. Kanobe et al. [56] concluded that aphids reduced the levels of PUFAs in the leaves and seeds of soybean plants. Aphids appear to affect the activity of some desaturases that are responsible for converting oleate (18:1) into linoleic acid (18:2) and ALA (18:3). The conversion takes place in chloroplasts, and the main desaturase involved is FAD 6 [61]. In addition to affecting desaturase activity, it has been proposed that aphids also indirectly influence the levels of PUFAs on the activity of KAS II. This is involved in the elongation of palmitate (16:0) to produce stearate, which by desaturation would lead to the formation of oleate [56]. Li et al. [40] confirmed that some FADs are important susceptibility factors in plant–aphid interactions, and that green peach aphid *Myzus persicae* resistance is more strongly associated with differences in the abundances C 18:1 and C18:2 compared to the abundance of C 16 in *Arabidopsis thaliana*. Limiting the amount of ALA and linoleic acid, aphids may limit the ability of plants to produce volatile compounds that would not only adversely affect the pests’ performance directly, but would also attract aphid predators and parasitoids. Our results are consistent with the putative effect of aphids on the levels of PUFAs (oxylipin precursors), thereby decreasing the levels of PhytoFs and PhytoPs. Furthermore, PhytoFs levels may be enhanced by higher water content and higher oxygen pressure, giving rise to the oxidation conditions required for the synthesis of PhytoFs. Moreover, we detected that wheat samples treated with *Rhopalosiphum padi* aphids had lower levels of PhytoFs than samples treated with *Sitobion avenae*, indicating that the former could reduce the levels of PUFAs more than the latter. Therefore, PhytoFs could be postulated to be an early indicator of aphid hosting of this plant species. Moreover, this is the first report on the quantification of non-enzymatic PhytoFs and PhytoPs in wheat leaves in their response to aphids.

### Strengths and Limitations of the Study

Our study reveals the relationship between aphid invasion and the response generated in the wheat plant through the non-enzymatic oxidative stress markers 9-F_1t_-PhytoP, 9-*epi*-9-F_1t_-PhytoP, *ent*-16-F_1t_-PhytoP + *ent*-16-*epi*-16-F_1t_-PhytoP), *ent*-9(*RS*)-12-*epi*-ST-Δ^10^-13-PhytoF, *ent*-*1*6(RS)-13-*epi*-ST-Δ^14^-9-PhytoF and *ent*-16(*RS*)-9-*epi*-ST-Δ^14^-10-PhytoF, which has never described before as these markers are not commercially available. Although our study has as a limitation in the low number of samples in the control group (6 samples), we found that PhytoP and PhytoF levels in wheat leaves depend on aphid hosting. Our findings indicated that the levels of *ent*-16(*RS*)-9-*epi*-ST-Δ^14^-10-PhytoF were higher in *Sitobion avenae* than in *Rhopalosiphum padi*; therefore, the mechanism by which the aphids decreased the level of phytofurans in the wheat leaves must be studied. Furthermore, the determination of gene expression or other types of oxidative stress markers, such as malondialdehyde as an indicator of ROS accumulation, antioxidant enzymes and chlorophyll, could be interesting to complete our findings. This will lead to more conclusive results, and a more global view of the influence of aphid attack on oxidative stress markers in wheat leaves.

## 4. Materials and Methods

### 4.1. Wheat Samples

The experimental procedure is fully explained elsewhere [5]. Briefly, wheat (cv Pedrosa) plants were grown in 9 cm × 9 cm × 10 cm pots with vermiculite (Asfaltex S.A., Barcelona, Spain) in growth chambers. A factorial combining two [CO_2_] (ambient: 400 ppm, elevated: 700 ppm) and two nitrogen rates returned four treatments. Eight days after sowing (DAS), two nitrogen treatments were established in which plants were watered with either full Hoagland solution (high nitrogen), or with Hoagland solution where the nitrogen was reduced to 20% of the full solution (low nitrogen); the nutrient solution was applied three times a week in both treatments.

The day:night cycles were 14:10 h, with three Philips Green Power LED Production Modules Deep Red/Blue 150 providing 200 µmol m^−2^ s^−1^ at the canopy level. The daytime temperature was 20.7 ± 0.01 °C, and the nighttime temperature was 6.2 ± 0.02 °C; the vapor pressure deficits were 0.53 ± 0.005 kPa and 0.27 ± 0.003 kPa, respectively. A feeding behavior assay was initiated 28 DAS, at the onset of detectable effects of [CO_2_] and nitrogen on wheat plants. *Rhopalosiphum padi* and *Sitobion avenae* were allowed to feed on the youngest fully expanded leaves of the test plants for 8 h, once they were connected to the EPG device (EPG Systems, Wageningen, The Netherlands). The leaves were frozen (−80 °C) immediately after finalizing these assays, and used for the phytoprostanes and phytofurans analysis (next section). A total of 29 leaf samples were included in this study; 12 were treated with *Sitobion avenae*, 11 with *Rhopalosiphum padi*, and 6 were controls with no aphids. A larger sample of aphid-treated leaves was used to account for the larger variability associated with single-aphid treatments.

### 4.2. Standards and Reagents

LC–MS-grade solvents (methanol, water, formic acid and acetonitrile) were purchased from J.T. Baker (Phillipsburg, NJ, USA). Hexane was obtained from Panreac (Castellar del Vallés, Barcelona, Spain). Bis–Tris (bis(2-hydroxyethyl)amino-tris(hydroxymethyl)methane) was purchased from Sigma–Aldrich (St. Louis, MO, USA). Sodium hydroxide (NaOH), and phosphoric acid were acquired from PanReac Química (Barcelona, Spain). The SPE cartridges used were Strata cartridges (Strata X-AW, 100 mg/3 mL), which were acquired from Phenomenex (Torrance, CA, USA).

PhytoPs, including 9-F_1t_-PhytoP; *ent*-16-F_1t_-PhytoP; *ent*-16-*epi*-16-F_1t_-PhytoP; 9-*epi*-9-F_1t_-PhytoP; 9-D_1t_-PhytoP; 9-*epi*-9-D_1t_-PhytoP; 16-B_1_-PhytoP; and 9-L_1_-PhytoP, as well as the PhytoFs *ent*-16(*RS*)-9-*epi*-ST-Δ^14^-10-PhytoF; *ent*-9(*RS*)-12-*epi*-ST-Δ^10^-13-PhytoF; and *ent*-16(*RS*)-13-*epi*-ST-Δ^14^-9-PhytoF, were synthesized by Durand’s team at the Institut des Biomolecules Max Mosseron (IBMM) (Montpellier, France) [19] (see Figure 3). The synthetic isoprostane 8-iso-PGF_2α_-d_4_ (containing four deuterium atoms at positions 3, 3′, 4 and 4′), used as the internal standard, was purchased from Cayman Chemicals (Ann Arbor, MI, USA).

### 4.3. Stock, Working and Standard Solutions

Individual stock solutions of 9-F_1t_-PhytoP, 9-*epi*-9-F_1t_-PhytoP, 9-L_1_-PhytoP, 16-B_1_-PhytoP, *ent*-16-F_1t_-PhytoP and *ent*-16-*epi*-16-F_1t_-PhytoP; *ent*-16(*RS*)-9-*epi*-ST-Δ^14^-10-PhytoF, *ent*-9(*RS*)-12-*epi*-ST-Δ^10^-13-PhytoF and *ent*-16(*RS*)-13-*epi*-ST-Δ^14^-9-PhytoF were prepared in MeOH:H_2_O (50:50, *v*/*v*) at a concentration of 1000 nM. Multi-component working solutions were obtained via dilution of the individual stock solutions in H_2_O:MeOH (50:50; *v*/*v*) to produce the standard solutions as follows: 1000, 500, 250, 125, 62.5, 31.2, 15.6, 7.8, 3.9, 1.9, 0.95 and 0.47 nM, which were required for the calibration curve.

### 4.4. Phytoprostanes and Phytofurans Analysis

The PhytoPs and PhytoFs in wheat leaves were extracted following the protocol described by Collado-González et al. and Domínguez-Perles et al. [31,32], with minor modifications. Briefly, 2-g samples were grinded in a mortar with 5 mL of methanolic butylated hydroxyanisole (BHA) (99.9:0.1, *v*/*w*). The extracts were centrifuged at 2000× *g* for 10 min, and the supernatants were collected and cleaned up using solid-phase extraction (SPE) with Strata X-AW cartridges (Phenomenex, Torrance, CA, USA), according to the procedure described [34].

The PhytoPs and PhytoFs were separated chromatographically with a UHPLC coupled with a 6460 triple quadrupole-MS/MS (Agilent Technologies, Waldbronn, Germany), using the analytical column BEH C_18_ (2.1 mm × 50 mm, 1.7 μm) (Waters, Milford, MI, USA). The column temperatures were 6 °C (both left and right). The mobile phases consisted of Milli-Q-water/acetic acid (99.99:0.01, *v*/*v*) (A) and methanol/acetic acid (99.99:0.01, *v*/*v*) (B). The injection volume and flow rate were 20 μL and 0.2 mL min^–1^, respectively, upon the following linear gradient (time (min), % B): (0.00, 60.0%); (2.00, 62.0%); (4.00, 62.5%); (8.00, 65.0%); and (8.01, 60.0%). An additional post-run of 1.5 min was considered for column equilibration. The spectrometric analysis was conducted in multiple reaction monitoring mode (MRM) operated in negative mode, assigning preferential MRM transition for the corresponding analytes. The ionization and fragmentation conditions were as follows: gas temperature 325 °C, gas flow 8 L min^–1^, nebulizer 30 psi, sheath gas temperature 350 °C, jet stream gas flow 12 L min^–1^, capillary voltage 3000 V, and nozzle voltage 1750 V, according to the most abundant product ions. The data acquisition and processing were performed using MassHunter software version B.04.00 (Agilent Technologies). The quantification of the PhytoPs and PhytoFs detected in the plant samples was performed using authentic standards, according to standard curves that were freshly prepared as described in the previous section. The selected reaction monitoring and chemical names used were according to the nomenclature system of [62]. The acquisition parameters are summarized in Table 4.

### 4.5. Statistical Analysis

The UHPLC–MS/MS data were acquired and processed using MassHunter software version B.04.00 from Agilent Technologies. Values below the limit of quantification (LOQ) were set at 0.5 LOQ. Further data analysis was carried out in MATLAB R2019b (MathWorks, Inc., Natick, MA, USA), and PLS Toolbox 8.0 (Eigenvector Research, Inc., Wenatchee, WA, USA) was used for principal component analysis (PCA) and partial least square discriminant analysis (PLSDA) models of the autoscaled data. The receiver operating characteristic (ROC) curves were based on the PLSDA models. Significance of models was tested using permutation testing (500 permutations).

## 5. Conclusions

In summary, we obtained PhytoP and PhytoF fingerprints of wheat samples, with and without *Rhopalosiphum padi* and *Sitobion avena* aphids invasion, making this study the first report to quantify non-enzymatic PhytoFs and PhytoPs in wheat leaves in response to aphids. Our results reveal that the levels of PhytoFs were much higher than those of PhytoPs, and that these values decreased under aphid hosting. We demonstrated that the levels of PhytoPs and PhytoFs depend on the infestation of the wheat plants, and that they were not influenced by high or low CO_2_/N regimes applied, resulting in the postulation that PhytoFs are strong indicators of aphid hosting in wheat leaves.

We suggested that aphids may have a strong effect on levels of PUFAs (particularly ALA)–oxylipin precursors, decreasing the levels of PhytoFs and PhytoPs when plants are infested with aphids; however, further studies are required.

## Figures and Tables

**Figure 1 molecules-28-04133-f001:**
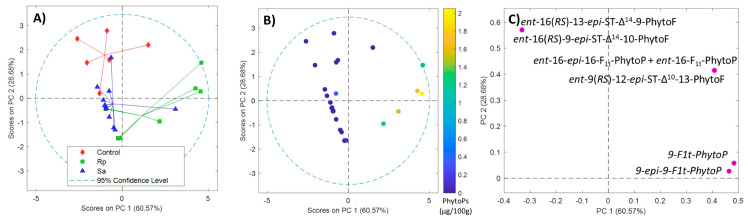
PCA results from the analysis of PhytoPs and PhytoFs in wheat samples. Score plots of PC1 vs. PC2 (**A**); score plots of PC1 vs. PC2 colored by PhytoPs content (**B**); and loadings plots of PC1 vs. PC2 (**C**). Note: *Rhopalosiphum padi* (Ra); *Sitobion avenae* (Sa).

**Figure 2 molecules-28-04133-f002:**
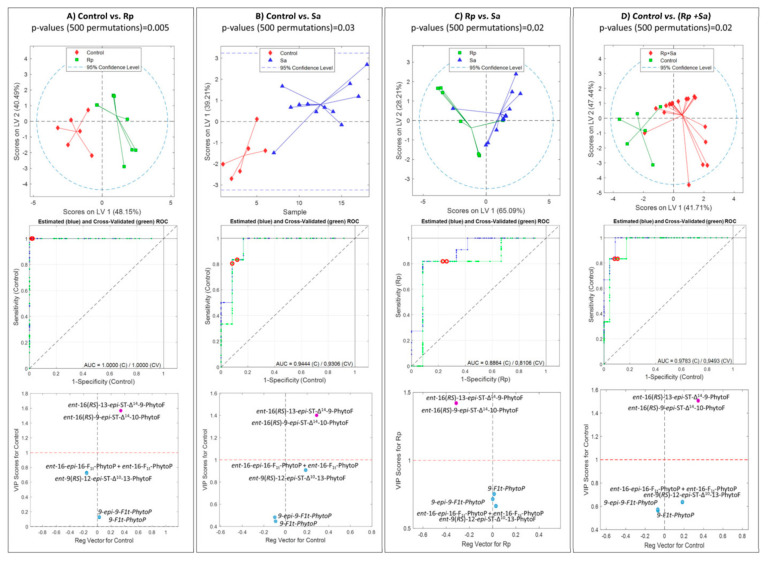
Discrimination between wheat leaves with *Rhopalosiphum padi* (Rp), *Sitobion avenae* (Sa) and controls with no aphids. PLSDA (top), ROC curves (middle) and VIP scores versus regression vectors (bottom) from PLSDA. (**A**) control vs. Rp; (**B**) control vs. Sa; (**C**) Rp vs. Sa; and (**D**) control vs. (Rp + Sa). Blue lines represent estimated PLSDA ROC curves (calibration set); green lines represent estimated PLSDA ROC curves (cross-validation); dashed lines represent 50% lines; and circles indicate model thresholds. PLSDA, partial least square discriminant analysis; ROC, receiver operating characteristic; VIP, variable importance in projection.

**Figure 3 molecules-28-04133-f003:**
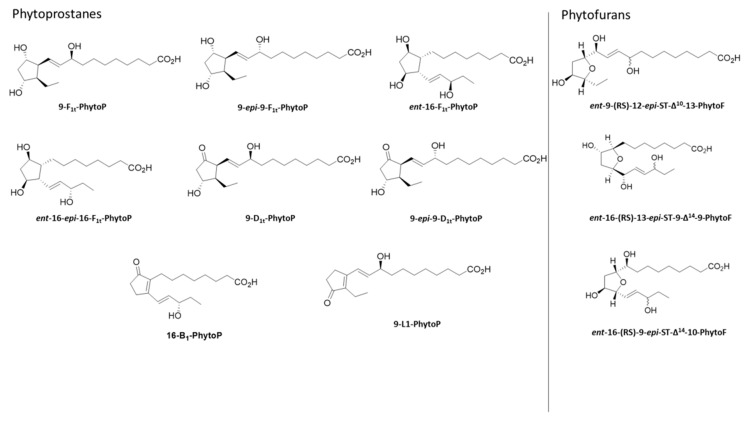
Chemical structures of phytoprostanes and phytofurans.

**Table 1 molecules-28-04133-t001:** Median and interquartile ranges (IQR) (25–75%) of phytoprostane and phytofuran concentrations in wheat leaves challenged with *Rhopalosiphum padi* (Rp) and *Sitobion avenae* (Sa), and controls with no aphids.

		Control		*Rhopalosiphum padi*		*Sitobion avenae*		*p*-Value			
	Compound	Median (µg/100 g)	IQR (µg/100 g)	Median (µg/100 g)	IQR (µg/100 g)	Median (µg/100 g)	IQR (µg/100 g)	Control vs. Rp	Control vs. Sa	Rp vs. Sa	Control vs. Rp + Sa
PhytoP	9-F_1t_-PhytoP	0.0002	0.0002–0.005	0.01	0.0011–0.4	0.0002	0.0002–0.012	0.05	0.4	0.16	0.13
	9-*epi*-9-F_1t_-PhytoP	0.0002	0.0002–0.0002	0.012	0.0002–0.9	0.009	0.0002–0.02	0.12	0.10	0.5	0.09
	*ent*-16-*epi*-16-F_1t_-PhytoP + *ent*-16-F_1t_-PhytoP	0.003	0.0002–0.007	0.0002	0.0002–0.008	0.0002	0.0002–0.002	0.80	0.20	0.20	0.30
PhytoF	*ent*-9(*RS*)-12-*epi*-ST-Δ^10^-13-PhytoF	4	0.0002–11	0.0002	0.0002–12	0.0002	0.0002–0.002	0.80	0.20	0.20	0.30
	*ent*-16(*RS*)-13-*epi*-ST-Δ^14^-9-PhytoF	3	2–4	0.0002	0.0002–0.0002	1	0.7–2	0.0002	0.010	0.0008	0.0011
	*ent*-16(*RS*)-9-*epi*-ST-Δ^14^-10-PhytoF	7	5–8	0.0002	0.0002–0.0002	2	1.7–4	0.0002	0.010	0.0008	0.0011
Total	PhytoP	0.007	0.005–0.02	0.01	0.005–1.2	0.02	0.004–0.03	0.40	0.50	0.50	0.40
	PhytoF	16	11–23	4	0.0002–12	5	4–7	0.02	0.005	0.2	0.005

**Table 2 molecules-28-04133-t002:** Comparison of phytoprostane profiles in our study and published reports.

Plant/Sample	9-F_1t_-PhytoP	9-*epi*-9-F_1t_-PhytoP	9-D_1t_-PhytoP	9-*epi*-9-D_1t_-PhytoP	9-L_1_-PhytoP	16-B_1_-PhytoP	*ent*-16-F_1t_-PhytoP + *ent*-16-*epi*-16-F_1t_-PhytoP	Reference
Wheat Leaves	✓	✓					✓	This Study
*Cucumis melo* L. leaves	✓	✓			✓	✓	✓	[41]
Date tree leaves	✓	✓	✓	✓	✓	✓		[42,43]
Chilean hazelnut (*Gevuina avellana* Mol., Proteaceae) cotyledons	✓	✓	✓	✓		✓	✓	[46]
Macroalgae	✓	✓			✓	✓		[44]
Brown macroalgae (*Ectocarpus siliculosus*)	✓	✓			✓	✓	✓	[45]
Brown macroalgae (*Laminaria digitate*)	✓	✓			✓	✓	✓	[45]
Brown macroalgae (*Fucus spiralis*)	✓	✓			✓	✓	✓	[45]
Red macroalgae (*Osmundea pinnatifida*)	✓	✓					✓	[45]
Red macroalgae (*Grateloupia turuturu*)	✓	✓					✓	[45]
Brown macroalgae (*Pelvetia canaliculata*)	✓	✓			✓	✓	✓	[45]
*Passiflora edulis* Sims shell	✓	✓	✓	✓	✓	✓		[18]
*Passiflora tripartita* var.mollisima shell	✓	✓	✓	✓	✓	✓	✓	[47]
*Physalis peruviana* calyx	✓	✓	✓	✓	✓	✓	✓	[48]

✓ means detection of the compound in the corresponding food or plant sample.

**Table 3 molecules-28-04133-t003:** Comparison of phytofuran profiles in our study and published reports.

Plant/Food Sample	*ent*-16(*RS*)-9-*epi*-ST-Δ^14^-10-PhytoF	*ent*-9(*RS*)-12-*epi*-ST-Δ^10^-13-PhytoF	*ent*-16(*RS*)-13-*epi*-ST-Δ^14^-9-PhytoF	Reference
Wheat Leaves	✓	✓	✓	This Study
*Cucumis melo* L. leaves	✓	✓		[41]
Date tree leaves	✓	✓	✓	[42,43]
Chilean hazelnut (*Gevuina avellana* Mol., Proteaceae) cotyledons	✓	✓	✓	[46]
Brown macroalgae (*Ectocarpus siliculosus*)	✓	✓	✓	[45]
Brown macroalgae (*Laminaria digitate*)	✓	✓	✓	[45]
Brown macroalgae (*Pelvetia canaliculata*)	✓			[45]
Red macroalgae (*Osmundea pinnatifida*)			✓	[45]
Red macroalgae (*Grateloupia turuturu*)	✓		✓	[45]
Brown macroalage (*Fucus spiralis*)	✓	✓	✓	[45]

✓ means detection of the compound in the corresponding food or plant sample.

**Table 4 molecules-28-04133-t004:** Acquisition parameters and main figures of merit of the LC–MS/MS method. The ESI mode was negative in all cases.

Compound		RT (min)	MRM Transition (*m*/*z*)	Fragmentor (V)	CE (V)
Phytoprostanes	*ent*-16-*epi*-16-F_1t_-PhytoP	1.583	327.1 > 283.2	80	15
			327.1 > 225.1	80	15
	9-F_1t_-PhytoP	1.631	327.2 > 273.1	110	15
			327.2 > 171.0	110	15
	*ent*-16-F_1t_-PhytoP	1.712	327.2 > 283.2	80	10
			327.2 > 225.1	80	10
	9-*epi*-9-F_1t_-PhytoP	1.785	327.2 > 272.8	110	10
			327.2 > 171.0	110	10
	9-D_1t_-PhytoP	1.791	325.2 > 307.3	100	4
			325.2 > 134.7	100	4
	9-*epi*-9-D_1t_-PhytoP	2.022	325.2 > 307.2	100	7
			325.2 > 134.9	100	7
	16-B_1_-PhytoP	2.62	307.2 > 223.2	100	10
			307.2 > 235.1	100	10
	9-L_1_-PhytoP	3.079	307.2 > 185.1	110	7
			307.2 > 185.2	110	7
Phytofurans	*ent*-9-(RS)-12-*epi*-ST-Δ^10^-13-PhtoF	0.906	344.0 > 300.0	110	10
			344.0 > 255.9	110	10
	*ent*-16-(RS)-9-*epi*-ST-Δ^14^-10-PhytoF	1.501	343.9 > 209.9	90	12
			343.9 > 201.1	90	12
	*ent*-16-(RS)-13-*epi*-ST-Δ^14^-9-PhytoF	1.523	343.0 > 171.1	90	22
			343.0 > 97.2	90	22

Note: RT, retention time; CE, collision energy.

## Data Availability

All data presented in the study are available on request from the corresponding authors (angelgil@cebas.csic.es; victor.sadras@sa.gov.au).

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
