# Peer review of "Wheat Oxylipins in Response to Aphids, CO2 and Nitrogen Regimes"

_molecules, 2023, doi:10.3390/molecules28104133_

Round 1

Reviewer 1 Report

In the manuscript titled Wheat oxylipins in response to aphids, CO2 and nitrogen regimes, the authors found that PhytoFs could be postulated as an early indicator of aphid hosting of wheat plants. Generally the experiments were well-designed. Below I have some small questions and I listed below:

1. Please check the language description, such as line 69 “Reactive Oxigen Species (ROS)” changes to reactive oxygen species (ROS), etc.

2. Are there phenotypic changes among treatments? If possible, please provide phenotypic pictures.

3. In fact, it is easy to determinate the ROS accumulation, such as MDA, H2O2.

4. Deepen the research significance of this article in discussion.

Author Response

Please, see file attached

Reviewer 2 Report

Reviewer comments

Manuscript: molecules-2232597 - Wheat oxylipins in response to aphids, CO2 and nitrogen regimes

The authors aimed to determine whether aphids feeding on wheat could trigger a defensive plant reaction to oxidative stress involving plant oxylipins. With a focus on this wheat-aphid system, authors reported the role of carbohydrates, with emphasis on osmotic stress, on the fitness and behavior of these insects. Using the same experimental setting, authors investigated whether aphids trigger a defensive reaction of the plant in terms of oxidative stress related to the generation of oxylipins. Both Rhopalosiphum padi and Sitobion avenae reduced the concentration of ent-16(RS)-13-epi-ST-Δ 14-9- PhytoF and ent-16(RS)-9-epi-ST-Δ 14-10-PhytoF in relation to controls with little or no effect on PhytoPs. The authors results are consistent with aphids affecting the levels of PUFAs, oxylipin precursors, decreasing the levels of PhytoFs in wheat leaf. Therefore, PhytoFs could be postulated as an early indicator of aphid hosting of this plant species. This is the first report on the quantification of non-enzymatic PhytoFs and PhytoPs in wheat leaves in response to aphids.

The English of the text is well written and well readable.

English language and style are fine and minor spell check required.

The uniqueness of the text is more than 90% by AntiPlagiarism.NET.

The text contains some misspellings and typos.

There are some comments and questions:

1) Line 58 - synthetized - should be - synthesized.

2) Line 137 - counteprarts - should be - counterparts.

3) Line 153 - diferences - should be - differences

4) Line 226 - gridning - should be - grinding.

5) Line 241 - Lycopersicum sculentum - should be - Lycopersicum esculentum.

6) Line 315 - complet - should be  - complete.

7) Line 385 - chromatographicaly -should be- chromatographically.

8) Why didn't the authors do the analysis of viruses in aphids? Maybe not only aphids effect on wheat.

9) Why do the authors check PhytoPs and PhytoFs in wheat? Following manuscripts demonstrate many markers for stress measure in plants.

Paes De Melo, B., Carpinetti, P.A., Fraga, O.T., Rodrigues-Silva, P.L., Fioresi, V.S., De Camargos, L.F., Ferreira, M., 2022. Abiotic Stresses in Plants and Their Markers: A Practice View of Plant Stress Responses and Programmed Cell Death Mechanisms. Plants (Basel). 11 (9). doi: 10.3390/plants11091100.

Akter, S., Khan, M.S., Smith, E.N., Flashman, E., 2021. Measuring ROS and redox markers in plant cells. RSC Chem Biol. 2 (5), 1384-1401. doi: 10.1039/d1cb00071c. Explain and discuss it.

10) What is the mechanism of no effect on PhytoPs and decreasing the levels of PhytoFs in wheat leaf? I think the authors should check the expression of key genes in signaling pathways.

Please improve the manuscript according to the above comments.

Author Response

Please, see file attached.
